# Selectively driving cholinergic fibers optically in the thalamic reticular nucleus promotes sleep

**Kun-Ming Ni[1†], Xiao-Jun Hou[1,2†], Ci-Hang Yang[1], Ping Dong[1], Yue Li[1], Ying Zhang[1], Ping Jiang[1], Darwin K Berg[3], Shumin Duan[1], Xiao-Ming Li[1,4*]**

[1]Department of Neurobiology, Institute of Neuroscience, Key Laboratory of Medical Neurobiology of the Ministry of Health of China, Collaborative Innovation Center for Brain Science, Zhejiang University School of Medicine, Hangzhou, China; [2]Fuzhou Children's Hospital, Fujian, China; [3]Neurobiology Section, Division of Biological Sciences, Center for Neural Circuits and Behavior, University of California, San Diego, La Jolla, United States; [4]Soft Matter Research Center, Zhejiang University, Hangzhou, China

**Abstract** Cholinergic projections from the basal forebrain and brainstem are thought to play important roles in rapid eye movement (REM) sleep and arousal. Using transgenic mice in which channelrhdopsin-2 is selectively expressed in cholinergic neurons, we show that optical stimulation of cholinergic inputs to the thalamic reticular nucleus (TRN) activates local GABAergic neurons to promote sleep and protect non-rapid eye movement (NREM) sleep. It does not affect REM sleep. Instead, direct activation of cholinergic input to the TRN shortens the time to sleep onset and generates spindle oscillations that correlate with NREM sleep. It does so by evoking excitatory postsynaptic currents via $\alpha$7-containing nicotinic acetylcholine receptors and inducing bursts of action potentials in local GABAergic neurons. These findings stand in sharp contrast to previous reports of cholinergic activity driving arousal. Our results provide new insight into the mechanisms controlling sleep.

**\*For correspondence:** lixm@zju.edu.cn

[†]These authors contributed equally to this work

## Introduction

Cholinergic neurons in the basal forebrain and brainstem provide widespread projections to numerous brain regions, including the cerebral cortex, hippocampus, thalamus, and TRN (*Mesulam et al., 1983*). Abundant behavioral and electrophysiological data suggest that cholinergic neurons play critical roles in regulating paradoxical sleep and the sleep-waking cycle (*Koyama et al., 1994*; *Greco et al., 1999*; *Bellingham and Funk, 2000*; *Kobayashi et al., 2003*; *Boutrel and Koob, 2004*; *Darbra et al., 2004*; *Brown et al., 2012*), as well as in promoting arousal. Recently, it has been shown that selective activation of cholinergic neurons in the basal forebrain induces immediate transition from sleep to waking (*Han et al., 2014*).

The TRN serves as a gate for information flow between the cerebral cortex and thalamus due, in part, to its location near thalamic white matter and to the large numbers of GABAergic neurons it contains (*Houser et al., 1980*; *Pinault, 2004*). The TRN is important for supporting attention (*McAlonan et al., 2008*), sleep regulation (*Steriade et al., 1987*; *Espinosa et al., 2008*), and sleep-dependent memory consolidation (*Fogel and Smith, 2011*). The physiological effects of cholinergic projections to the TRN, however, remain unclear.

In the present study, we took the advantage of the temporal precision and cell-type specificity of optogenetics to show that optical activation of cholinergic input to the TRN stimulates local

**eLife digest** Sleep is one of the most familiar activities in our lives and yet there are still many unanswered questions related to how it is regulated. The cholinergic system (or the part of the nervous system that sends signals using a chemical called acetylcholine) is thought to be important for the phase of sleep that is most similar to being awake, so-called REM sleep. This collection of nerve cells has also been implicated in the process of waking up from sleep. However, it remains unclear how the cholinergic system acts on sleep.

Ni, Hou et al. have now used a technique called optogenetics to use light to stimulate the cholinergic system in specific areas in the brains of mice. These experiments found that the activation of the cholinergic system caused awake mice to fall asleep, and promoted more non-REM sleep in sleeping mice. As such, this discovery challenges the previously held view that cholinergic activity was linked to waking up.

Acetylcholine affects cells in a similar way to nicotine from cigarettes. In the future, Ni, Hou et al. would like to explore how many nicotine-like substances are released by the cholinergic system in specific brain areas, and to further investigate when and how sleep is promoted.

GABAergic neurons by activating α7-containing nicotinic cholinergic receptors (α7-nAChRs). The resulting increase in GABAergic activity in the TRN shortens sleep onset time and extends NREM sleep duration. These unexpected findings are quite different from those previously reported for cholinergic neurons in the basal forebrain and brainstem, which support arousal and REM sleep. The present results indicate that cholinergic projections to the TRN normally promote sleep onset and protect NREM sleep. The behavioral outcome of cholinergic activity across brain regions, therefore, depends on the combination of pathways activated and targets stimulated.

## Results

### Optical driving of cholinergic fibers in the TRN generates spindle oscillations

To research the function of cholinergic innervation in the TRN, we employed *ChAT-ChR2* mice in which channelrhodopsin2 (ChR2) was selectively expressed in cholinergic neurons under the control of acetylcholinesterase promotor (*Zhao et al., 2011*). A 473 nm laser was employed to activate the cholinergic fibers through a 200 μm optical fiber (*Figure 1A*). Histological imaging showed that the optical fiber was well implanted above the TRN (*Figure 1B*). We then tested EEG from the skull in vivo and found that optical stimulation of cholinergic fibers in the TRN could induce spindle oscillations both in waking and sleep states. To determine which stimulation intensity was most effective for inducing spindle oscillations, we used four intensities of 20 ms, 100 ms, 200 ms, and 400 ms pulses, respectively. The visual inspection revealed increasing EEG power with increasing stimulation intensity. However, when the pulse intensity increased to 200 ms, the EEG power nearly reached the maximum (*Figure 1C*). We also compared the spindle co-occurrence evoked by 20 ms, 100 ms, 200 ms, and 400 ms, respectively, during awake and NREM states. The co-occurrence of spindle oscillation was greater than that in the waking stage ($P < 0.01$; *Figure 1D*). Furthermore, spindle co-occurrence increased as the pulse duration increased ($P < 0.01$; *Figure 1D*). However, there was no significant difference in co-occurrence between the 200 ms and 400 ms stimulation pulse-evoked spindle oscillations (*Figure 1D*). Next, we compared the evoked and natural spindle oscillations and found there was no significant difference in their frequencies, amplitudes or durations (*Figure 1E*). These results demonstrate that the evoked spindle oscillation pattern mimicked the natural spindle oscillation pattern.

To investigate which stimulation frequency induced the highest spindle oscillation density, we used a 200-ms pulse with intervals of 1 s, 3 s, 6 s, or 8 s to drive the cholinergic fibers in the TRN, respectively (*Figure 1F*). We then quantified the spindle density and found that the light pulse with a 6 s interval evoked the highest spindle oscillation densities in both waking and NREM phases. The

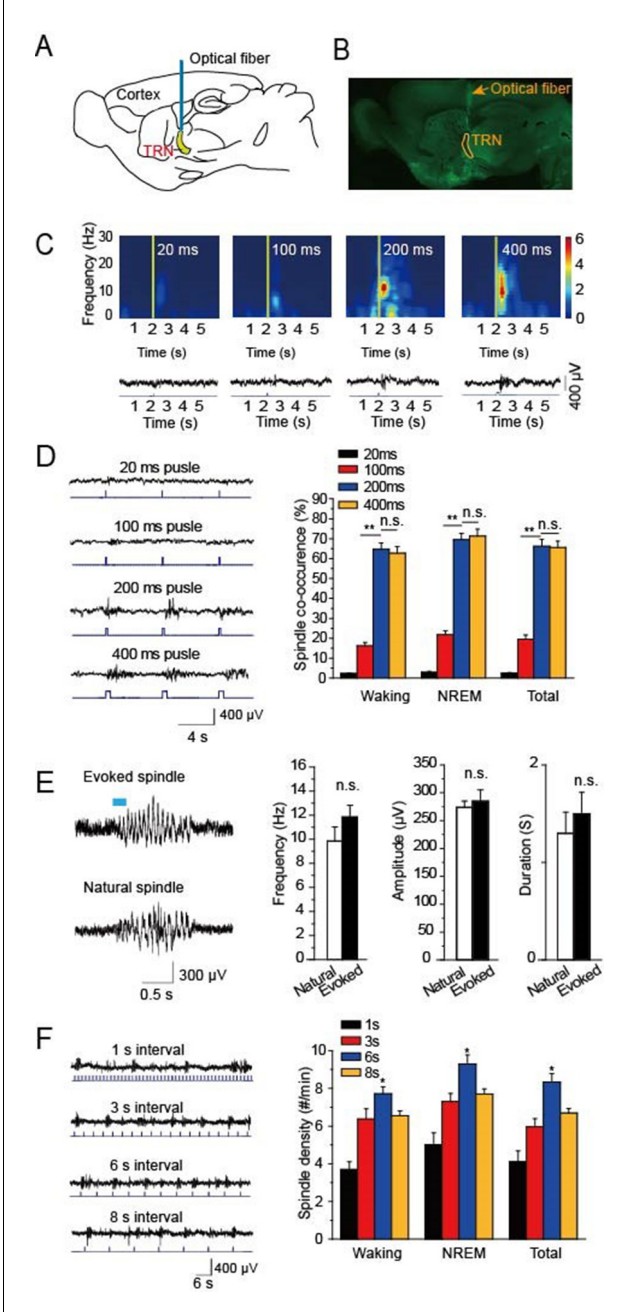

**Figure 1.** Optical driving of cholinergic fibers in the TRN generated spindle oscillations. (**A**) Placement of optical fiber above the TRN in *ChAT-ChR2* mice. (**B**) Histological imaging of implant location of optical fibers. (**C**) Upper, power spectrograms of EEG following stimulation for 20 ms, 100 ms, 200 ms, or 400 ms, respectively. Lower, representative spindle traces evoked by stimulation for 20 ms, 100 ms, 200 ms, or 400 ms, respectively. (**D**) Left, spindles evoked by different stimulation intensity. Right, average spindle co-occurrence rates during waking, NREM, and total recording periods. (**E**) Left, representative evoked spindle oscillation and natural spindle oscillation. Right, statistical analysis of spindle frequency, amplitude and duration. (**F**) Average spindle density induced by 200 ms stimulation of 1 s, 3 s, 6 s, 8 s intervals during waking, NREM, and total recording periods. Data represent mean ± SEM (n = 8 mice; **p < 0.01, two-tailed *t*-test between 200 ms stimulation and 400 ms stimulation or 100 ms stimulation).

evoked spindle density was 9.2 ± 0.5 per min in the NREM stage with a 6 s interval light pulse. These results suggested that a 6 s interval light pulse was most effective at evoking spindles.

## Activation of cholinergic fibers promotes sleep

To examine the roles of cholinergic input to the TRN in modulating sleep, *ChAT-ChR2* mice were raised under 12-hr light/12-hr dark conditions. The stimulation protocol applied 473 nm light (1.5 mW) for a 200-ms pulse at 6 s intervals for 1 hr, starting 4 hr after 'light off' (*Figure 2A*). After screening a range of conditions, this protocol appeared optimal and mimicked natural activity (*Figure 1*). Combined EEG, EMG, and video recordings were used to determine the sleep-waking patterns and to examine sleep architecture (*Figure 2B*) (*Lupi et al., 2008*). The stimulation protocol induced sleep onset in *ChAT-ChR2* mice (ChAT-ChR2-Sti) within 16 min, on average (*Figure 2C–E*). This was 50–60% quicker (*P* < 0.05) than in *ChAT-ChR2* mice without stimulation (ChAT-ChR2-Non-Sti) and in WT mice with stimulation (WT-Sti). There was no significant difference between WT-Sti mice and *ChAT-ChR2*-NonSti mice (*P* = 0.49). These results suggest that the shortened sleep onset was mediated by ChR2-induced activation of cholinergic fibers in the TRN. Correspondingly, the total sleep time of *ChAT-ChR2*-Sti mice stimulated for 1 hr increased significantly compared with that of both *ChAT-ChR2*-NonSti mice (*P* < 0.01) and WT-Sti mice (*P* < 0.01) (*Figure 2F*). The sleep architecture of *ChAT-ChR2*-Sti mice was also significantly different from that of either *ChAT-ChR2*-NonSti or WT-Sti mice. Time spent in NREM sleep in *ChAT-ChR2*-Sti mice increased by 160% compared with that of *ChAT-ChR2*-NonSti mice (*P* < 0.01), while no difference was seen in the duration of REM sleep (*Figure 2G*). No significant differences in sleep architecture were observed between *ChAT-ChR2*-NonSti and WT-Sti mice. To test how long it took for recovery of normal function, we recorded sleep states for 1 hr following cessation of stimulation, and found that it took an average of 40 ± 3 min to recover (*Figure 2—figure supplement 1*). These results suggest that activation of cholinergic fibers in the TRN can promote sleep onset and alter sleep architecture to protect sleep.

## Cholinergic-induced spindle oscillations correlate with NREM sleep

Previous studies revealed that spindle oscillations can protect and stabilize sleep (*Bové et al., 1994*; *Dang-Vu et al., 2010*; *Kim et al., 2012*). Accordingly, we examined the correlation between the density of cholinergic-induced spindle oscillations and the duration of NREM sleep over the recording period. Spindle density in *ChAT-ChR2*-Sti mice was significantly increased compared with that of *ChAT-ChR2*-NonSti and WT-Sti mice, while no differences were seen between *ChAT-ChR2*-NonSti and WT-Sti mice (p< 0.01, *Figure 3A*). In all three groups, spindle density was greatest (9.3 ± 0.5 min) during NREM sleep. Correlation analysis indicated a significant positive relationship between spindle oscillation density and NREM sleep duration (*Figure 3B*). These results suggest that evoked spindle oscillations can play roles similar to those of spontaneous spindle oscillations in protecting and stabilizing sleep.

## Cholinergic fibers modulate GABAergic neuron activity in the TRN

The TRN is thought to represent a shield surrounding the dorsal thalamus consisting of a homogenous population of parvalbumin (*PV*)-containing GABAergic neurons (*Houser et al., 1980*; *Ohara and Lieberman, 1985*). Confocal imaging of brain slices from B13 mice (which express GFP under a *PV* promoter) after immunostaining for *PV* confirmed that *PV*-positive neurons were abundant throughout the TRN (*Figure 4A*). To identify *PV*-positive neurons for mechanistic studies, we crossed *ChAT-ChR2* mice with B13 mice. This generated mice expressing both ChR2 in the cholinergic neurons and GFP in the *PV* neurons, permitting whole-cell recordings from GABAergic neurons while optically stimulating local cholinergic input. Immunostaining revealed numerous cholinergic fibers surrounding *PV*-positive neurons in the TRN of *ChAT-ChR2* mice (*Figure 4B*). First, we stimulated cholinergic neurons and found that a 5-ms pulse could induce the firing of action potentials in the cholinergic neurons when stimulating cell bodies (*Figure 4C*). We then tried 5-ms, 50-ms, 100-ms, and 200-ms light pulses to stimulate cholinergic fibers and recorded neuronal activity of GABAergic neurons in the TRN. (*Figure 4D*). We found that light pulses of short duration, such as 5 ms and 50 ms, failed to induce bursts of action potentials in GABAergic neurons. A light pulse of at least 100 ms was required to induce bursts of action potentials and spindles. Whole-cell recording in voltage-clamp mode from *PV* neurons in the TRN-containing brain slice showed that pure light

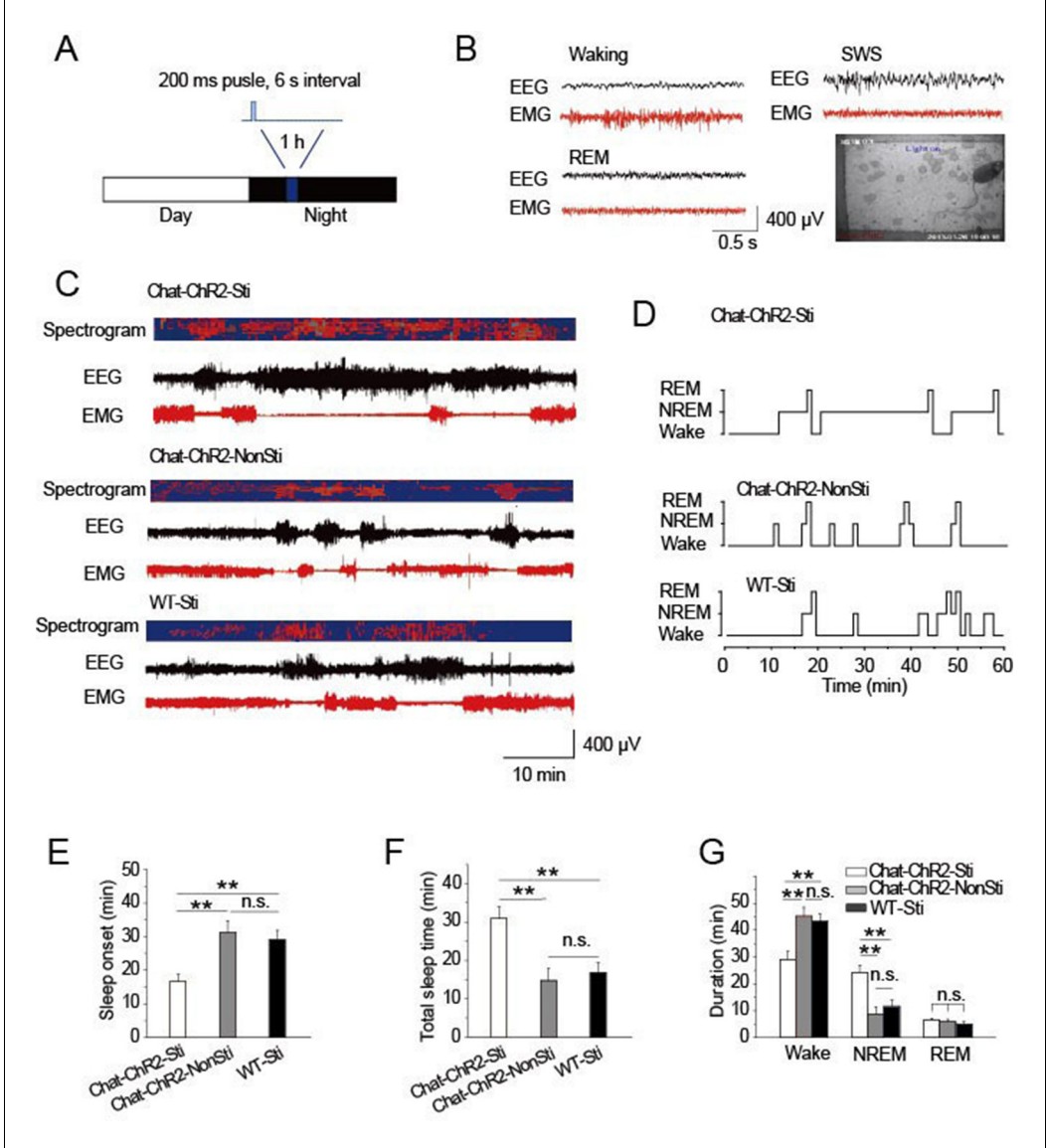

**Figure 2.** Activation of cholinergic fibers promoted sleep. (A) Stimulation protocol in the TRN of *ChAT-ChR2* mice. A 473 nm laser was used at 1.5 mW to deliver 200-ms pulses at 6 s intervals for 1 hr. (B) EEG, EMG, and video recordings were used to determine sleep-waking patterns and examine sleep architecture; examples of states (Waking, SWS, REM). (C) EEG trace during 1 hr stimulation period. (D) Representative hypnograms for *ChAT-ChR2*-Sti, *ChAT-ChR2*-NonSti and WT-Sti mice over 1 hr test period. Optical stimulation given to *ChAT-ChR2* and WT-Sti mice consisted of 200-ms pulses. (E) Latency to sleep in *ChAT-ChR2*-Sti, *ChAT-ChR2*-NonSti and WT-Sti mice. (F) Total sleep time of *ChAT-ChR2*-Sti, *ChAT-ChR2*-NonSti and WT-Sti mice during 1 hr stimulation. (G) Time spent in NREM stage for *ChAT-ChR2*-Sti, ChAT-ChR2-NonSti and WT-Sti mice. In (E, F, G), all data represent mean ± SEM (n = 8 mice, *p < 0.05, **p < 0.01, two-tailed *t*-test between *ChAT-ChR2*-Sti mice and *ChAT-ChR2*-NonSti or WT-Sti mice). See also *Figure 2— figure supplement 1*.

The following figure supplement is available for figure 2:

**Figure supplement 1.** Recovery time of normal function following cessation of stimulation.

stimulation (without injection current) of a 200-ms pulse evoked fast excitatory post-synaptic currents (EPSCs; *Figure 4E*, lower). They were accompanied by rhythmic high-frequency bursts of action potentials (APs) seen in current-clamp mode (*Figure 4E*, upper). In vivo extracellular recording in the TRN of anesthetized *ChAT-ChR2* mice revealed that the optical stimulation of cholinergic fibers induced spikes in the GABAergic neurons (*Figure 4F*). These results suggest that cholinergic projections to the TRN produce excitatory input sufficient to evoke bursts of APs in *PV*-containing TRN

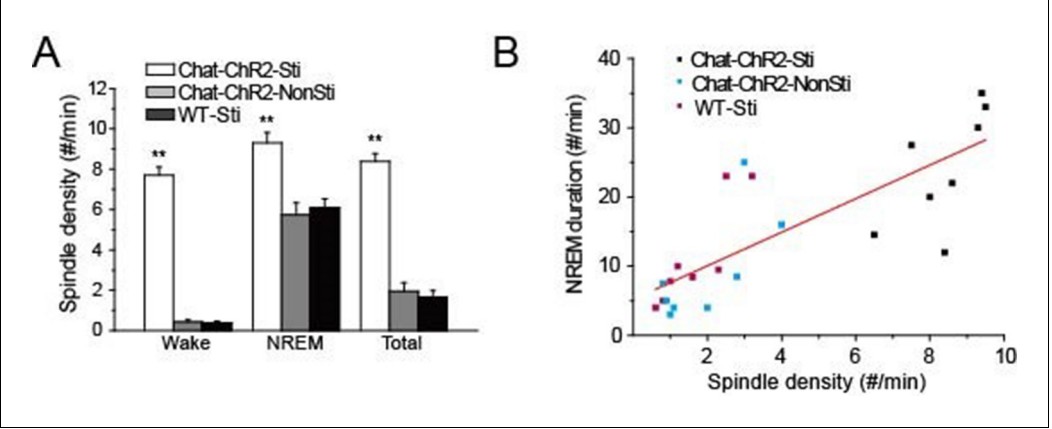

**Figure 3.** Correlation of spindle oscillation density and NREM sleep duration. (**A**) Spindle density of *ChAT-ChR2*-Sti, *ChAT-ChR2*-NonSti and WT-Sti mice during waking, NREM, and entire period (Total). Data represent mean ± SEM (n = 8 mice, *p < 0.05, **p < 0.01, two-tailed *t*-test between *ChAT-ChR2*-Sti mice and *ChAT-ChR2*-NonSti or WT-Sti mice). (**B**) Correlation shows a positive relationship between spindle oscillation density and NREM duration (n = 8).

neurons. To determine which cholinergic projections mediated these results, we injected AAV9-CAG-DIO-EGFP-2A-TVA and AAV9-CAG-DIO-RV-G into the TRN. Three weeks later, EnvA-pseudo-typed RV-ΔG-DsRed, which can infect efficiently and travel retrogradely, was injected into the same site. We found that the DsRed signal was detected in some cholinergic neurons of the nucleus basalis magnocellularis (nBM) in the basal forebrain (*Figure 4—figure supplement 1A*) and pedunculo-pontine tegmental nucleus (PPTg) of the brain stem (*Figure 4—figure supplement 1B*). Statistical results showed that about 30% of cholinergic neurons (VAChT antibody labeled) were positive for mCherry (retrograde virus labeled) in the nBM, and ~7% were positive for mCherry in the PPTg (*Figure 4—figure supplement 1C*). These results demonstrate that TRN neurons receive cholinergic projections from both the basal forebrain and brain stem.

## Light-evoked EPSCs in *PV*-containing TRN neurons are mediated by α7-nAChRs

To identify synaptic components mediating cholinergic-induced EPSCs in the TRN, we employed specific blockers. Recordings in the presence of tetrodotoxin (TTX, 1 µM) to block APs completely prevented optical stimulation of cholinergic fibers in the TRN of *ChAT-ChR2* mice from eliciting EPSCs in *PV*-containing neurons (*Figure 5A*). This is consistent with the EPSCs being AP-dependent, as expected for presynaptic stimulation of transmitter release. In contrast, light-evoked EPSCs were not blocked by including the combination of CNQX (20 µM) as an inhibitor of glutamatergic AMPA receptors, AP5 (50 µM) as an inhibitor of glutamatergic NMDA receptors, and picrotoxin (PTX, 100 µM) as an inhibitor of GABA$_A$ receptors (*Figure 5B*). This indicates that neither rapid glutamatergic nor GABAergic transmission is required for cholinergic induction of the EPSCs. Dihydro-β-erythroidine (DHβE), a selective antagonist of α4β2-containing nAChRs, decreased the amplitude of light-evoked EPSCs by 12%, 34%, and 99% at 5 nM, 10 nM, and 100 nM, respectively (*Figure 5C*). However, methyllycaconitine (MLA, 5 nM), a specific antagonist of α7-nAChRs, completely blocked the light-evoked EPSCs (*Figure 5D*) and the accompanying bursts of APs (*Figure 5E*). The concentration was consistent with previous research (*Bonfante-Cabarcas et al., 1996*). Immunostaining yielded extensive labeling for α7-nAChRs on *PV*-containing neurons in the TRN (*Figure 5F*). These results indicate that the light-evoked, AP-dependent release of ACh from cholinergic projections in the TRN elicits EPSCs and bursts of APs in GABAergic neurons of the TRN by activating α7-nAChRs.

We found that a light pulse of at least 100 ms duration could induce bursts of action potentials in GABAergic neurons in the TRN (*Figure 4*). It might be that the activation of GABAergic neurons by cholinergic fiber stimulation was an indirect effect via α7 cholinergic receptors (*Figure 5*). It requires a certain amount of Ach release to excite postsynaptic GABAergic neurons to fire action potentials

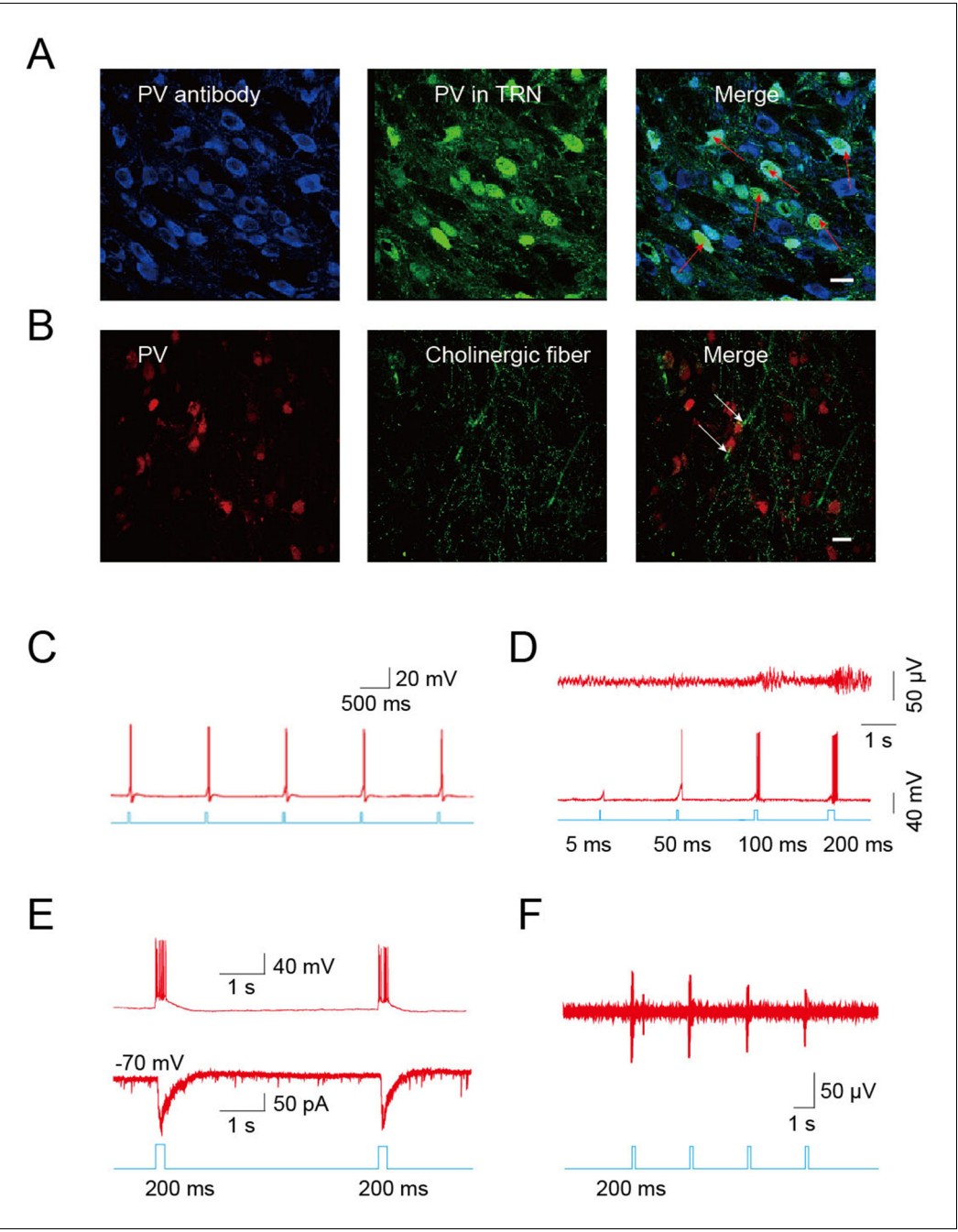

**Figure 4.** Cholinergic fibers excited GABAergic neurons in the TRN. (A) Confocal imaging of brain slices from B13 mice showing densely distributed *PV*-positive *Figure 2—figure supplements 1A,3B,4A* neurons in the TRN. (B) Histological staining showing numerous cholinergic fibers surrounding *PV*-positive neurons in the TRN of *ChAT-ChR2* mice. In **A, B,** scale bar: 20 μm; n = 3 mice. (C) Action potentials in cholinergic neurons induced by 5-ms pulse stimulation. (D) Action potentials in GABAergic neurons evoked by 5 ms, 50 ms, 100 ms, and 200 ms stimulation of cholinergic fibers in TRN (bottom). Spindles could only be induced by 100 ms or 200 ms stimulation, but not by 5 ms or 50 ms (top). (E) Light pulse stimulation of 200 ms evoked EPSCs (bottom) and rhythmic high-frequency bursts of APs (top) in *PV* neurons of the TRN. (F) In vivo extracellular recording of the TRN in an anesthetized *ChAT-ChR2* mouse showing light-evoked spikes on GABAergic neurons. In **C, D, E, F,** n = 8 neurons of 3 mice. See also *Figure 4—figure supplement 1*.

The following figure supplement is available for figure 4:

**Figure supplement 1.** The TRN received cholinergic projections from both the basal forebrain and brain stem.

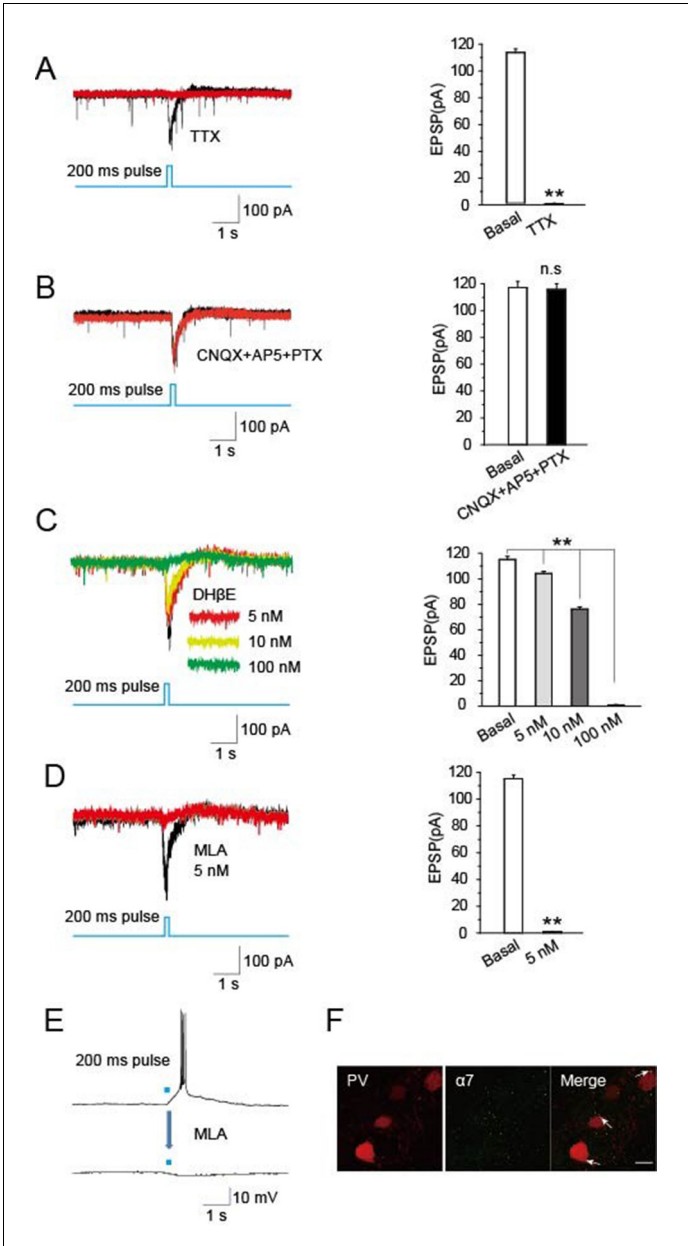

**Figure 5.** Optically induced EPSCs were blocked by MLA. (**A**) TTX (1 µM) blocked the induced EPSCs in *PV* neurons evoked by optical stimulation of cholinergic fibers in the TRN (n = 6). (**B**) CNQX (20 µM), AP5 (50 µM), and PTX (100 µM) together failed to block the induced EPSCs (n = 6). (**C**) DHβE (5, 10 nM) only partially blocked the evoked EPSCs. (**D,E**) MLA (5 nM) completely blocked the evoked EPSCs and bursts of APs (n = 6). For voltage clamp recording in (**A, B, C, D**), the membrane potential was held at -70 mV. (**F**) Immunostaining with an antibody specific for α7-nAChRs showed substantial levels of the receptors on *PV*-positive neurons. Scale bar: 20 µm (n = 4). Data represent mean ± SEM (*p < 0.05, **p < 0.01, one-way ANOVA was used).

and spindles. Light pulses of 10 ms or 50 ms are too short to induce the release of sufficient Ach, which evokes the burst of action potentials in postsynaptic GABAergic neurons and spindles.

## MLA in the TRN blocks cholinergic induction of sleep

Having found that α7-nAChRs mediate cholinergic input to *PV*-containing GABAergic neurons in the TRN, we tested whether this same pathway was responsible for cholinergic modulation of sleep onset and sleep protection. Half an hour before stimulation, we injected MLA (1 µl, 100 nM) or saline

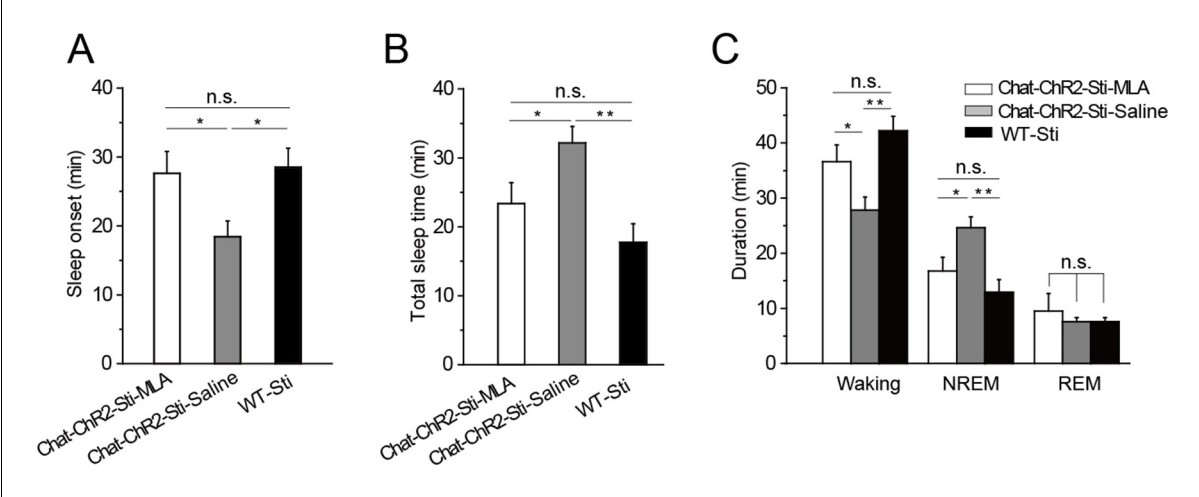

**Figure 6.** MLA blocked the decrease in sleep onset time and the increase in sleep duration induced by cholinergic activation. (**A**) Time to sleep onset for *ChAT-ChR2*-Sti-MLA, *ChAT-ChR2*-Sti-Saline, and WT-Sti mice. (**B**) Total sleep time of *ChAT-ChR2*-Sti, *ChAT-ChR2*-NonSti, and WT-Sti mice during 1 hr stimulation period. (**C**) Duration of waking, NREM, and REM for *ChAT-ChR2*-Sti-MLA, *ChAT-ChR2*-Sti-Saline and WT-Sti mice in 1 hr recording. Data represent mean ± SEM (n = 8 mice, *p < 0.05, **p < 0.01, two-tailed *t*-test between *ChAT-ChR2*-Sti-MLA mice and either *ChAT-ChR2*-Sti-Saline or WT-Sti mice). See also *Figure 6—figure supplement 1*.

The following figure supplement is available for figure 6:

**Figure supplement 1.** MLA decreased spontaneous sleep in the daytime.

(1 μl, 0.9% NaCl) locally into the TRN of *ChAT-ChR2* mice through a cannula implanted in the brain. The mice were then given optical stimulation (200-ms pulses at 6 s intervals, 1.5 mW blue laser) for 1 hr in the TRN. Quantification of the responses indicated that MLA completely blocked light-evoked changes in latency to sleep onset, total sleep time, and sleep architecture (*Figure 6*). None of the parameters were significantly different from those of WT-Sti mice. In contrast, the saline-injected mice could still be induced to sleep and showed increased NREM by the activation of cholinergic fibers during the 1 hr stimulation, compared with the *ChAT-ChR2*-Sti-MLA and WT-Sti groups. To further test MLA on spontaneous sleep, we injected saline and MLA into the TRN of *ChAT-ChR2* mice in the daytime, and found that in vivo injection of MLA (the antagonist of α7-nAChRs) significantly decreased spontaneous sleep time from 37.2 ± 4.1 min to 27.3 ± 3.2 min (p < 0.01, *Figure 6—figure supplement 1*). These results indicate that cholinergic promotion of sleep onset and sleep protection are mediated by activating *PV*-expressing neurons of the TRN through α7-nAChRs.

## Direct stimulation of GABAergic neurons in the TRN promotes sleep onset and alters sleep architecture

Having found that cholinergic input to *PV*-containing TRN neurons generates spindle oscillations and promotes NREM sleep via α7-nAChRs, we wanted to confirm whether direct stimulation of TRN GABAergic neurons could achieve the same effects. For this purpose, we used *VGAT-ChR2* mice in which ChR2 was specifically expressed in the GABAergic neurons (*Zhao et al., 2011*) so that those neurons would be selectively activated by light pulses delivered to the TRN. Testing the paradigm first in vitro showed that a light pulse of 10 ms applied to an acute slice could induce bursts of APs in GABAergic neurons of the TRN (*Figure 7A*). Testing three different optical stimulation patterns in vivo (20 Hz, 8 Hz, or single 10 ms pulses) indicated that 8 Hz stimulation was most successful at evoking spindle oscillations both in waking and NREM states (*P* < 0.05, *Figure 7B,C*). Accordingly, we used that stimulation protocol (10 ms of 8 Hz for 1 s at 6-s intervals) to induce sleep in *VGAT-ChR2* mice, which were raised under a 12-hr light/12-hr dark cycle (*Figure 7D*). Blue laser light of 473 nm was delivered into the TRN through a 200 μm optical fiber for 1 hr starting 4 hr after 'light off'. Both EEG and EMG were recorded, and sleep onset and sleep architecture were then

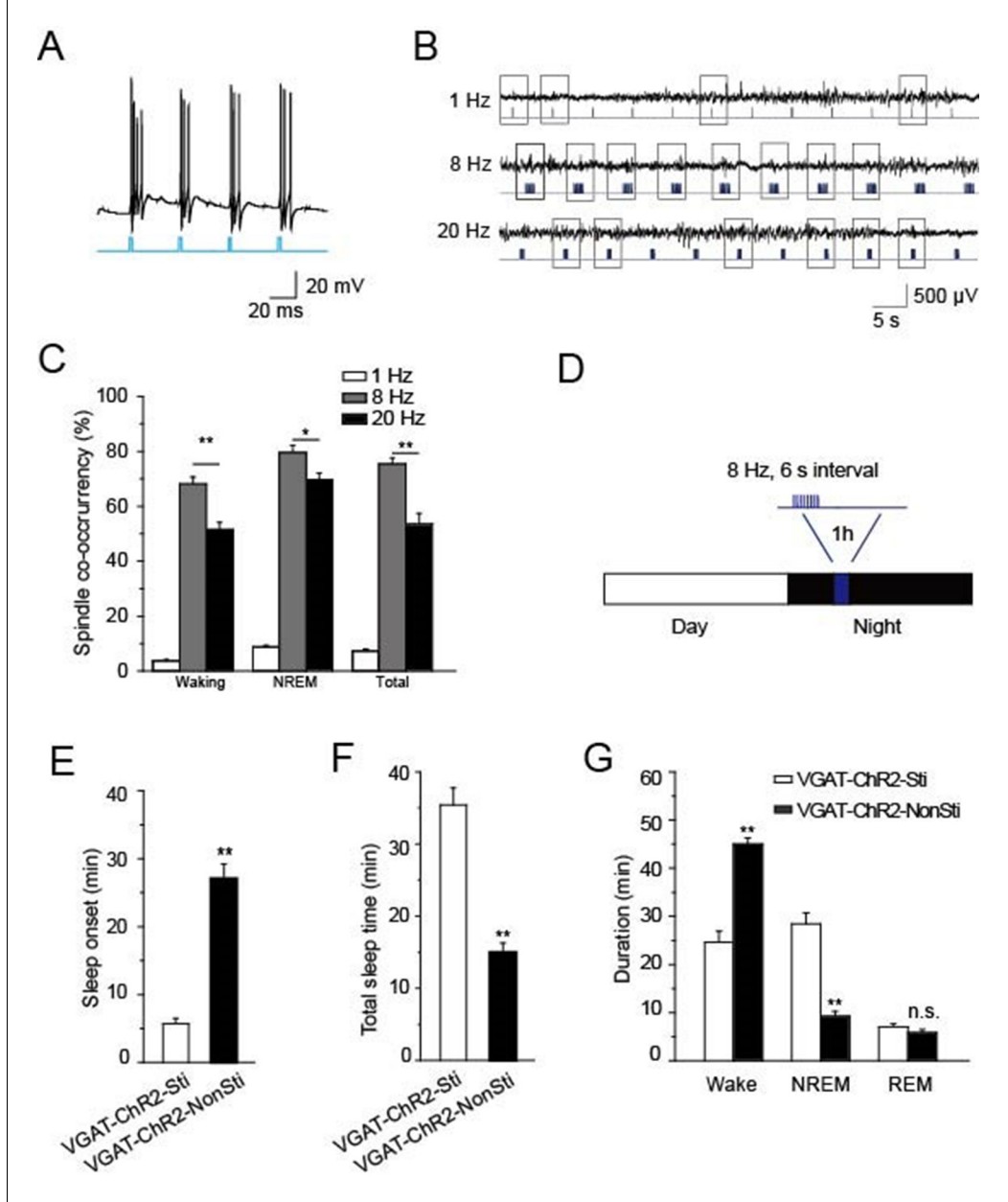

**Figure 7.** Direct stimulation of GABAergic neurons in the TRN promoted sleep onset and altered sleep architecture. (**A**) Direct optical drive (10-ms pulses, lower) induced bursts of APs in GABAergic neurons (upper). (**B**) Representative spindle oscillations evoked by different optical stimulation paradigms (1, 8, or 20 Hz). (**C**) Probability of spindle waves induced by 1 Hz, 8 Hz, and 20 Hz. (**D**) Stimulation protocol in the TRN of *VGAT-ChR2* mice. A 473 nm laser (1.5 mW) was given at 8 Hz for 1 s at 6-s intervals for 1 hr. (**E**) Sleep onset time of *VGAT-ChR2*-Sti and *VGAT-ChR2*-NonSti mice. (**F**) Total sleep time evoked by direct optical stimulation of GABAergic neurons in the TRN of *VGAT-ChR2* mice and unstimulated controls. (**G**) Durations of time in waking, NREM, and REM states for *VGAT-ChR2* mice with and without direct optical stimulation of GABAergic neurons in the TRN. Stimulation substantially decreased the duration of wake time, while increasing the duration of NREM time; it had no effect on REM time. In (**C, E, F, G**), data represent mean ± SEM (n = 7 mice, *p < 0.05, **p < 0.01, two-tailed *t*-test between *VGAT-ChR2* mice with spindle-like stimulation and *VGAT-ChR2* mice without stimulation).

measured. Optical stimulation significantly reduced the sleep onset time of *VGAT-ChR2* mice (*VGAT-ChR2*-Sti, 27.2 ± 3.2 min) compared with that of the controls (*VGAT-ChR2*-NonSti; 5.5 ± 0.8 min, *P* < 0.01, *Figure 7E*). Similarly, the duration of sleep during the 1 hr optical stimulation period also increased by 156% compared with that of the controls (*P* < 0.01, *Figure 7F*).

Furthermore, the architecture of sleep differed greatly in *VGAT-ChR2*-Sti mice vs. *VGAT-ChR2*-Non-Sti mice. Time spent in the NREM phase was significantly greater for *VGAT-ChR2*-Sti mice (27.2 ± 2.1 min) than for *VGAT-ChR2*-NonSti mice (15.2 ± 1.3 min, p < 0.01), while no significant differences were seen in the duration of the REM stage (*Figure 7G*). These results support that elevating the activity of GABAergic neurons in the TRN can promote sleep in ways that mimic the effects of cholinergic excitation of the TRN.

## Discussion

Our results show that cholinergic projections to the TRN can activate local GABAergic neurons via α7-nAChRs without requiring either glutamatergic or GABAergic rapid transmission. Furthermore, cholinergic fiber activation of GABAergic neurons in the TRN in vivo can promote sleep onset and increase the probability of spindle oscillation and duration of the NREM phase. Lastly, direct stimulation of GABAergic neurons in the TRN to elicit activity comparable to that induced by cholinergic input produces the same kinds of changes in sleep behavior seen following cholinergic activation of the neurons. These results demonstrate the capacity of cholinergic input, acting through GABAergic neurons in the TRN, to promote critical features of sleep.

Previous studies employing pharmacology, brain injury, and electrical stimulation revealed important contributions of cholinergic signaling to the regulation of the sleep-to-waking cycle and paradoxical sleep (*Campbell et al., 1969*; *Sarter et al., 2009*; *Brown et al., 2012*; *Hamlin et al., 2013*; *Sims et al., 2013*). These earlier studies suggested that cholinergic input to the TRN neurons was primarily inhibitory, acting on M2 muscarinic ACh receptors (*McCormick and Prince, 1986*). Local delivery of ACh to the TRN was seen to induce only a slow inhibitory response, leading to the conclusion that the ACh-induced bursts of APs observed were somehow the result of the GABAergic neurons in the TRN becoming hyperpolarized (*McCormick and Prince, 1986*). Instead, we showed cholinergic input to the TRN stimulates the GABAergic neurons via α7-nAChRs, and that this input through α7-nAChRs induces bursts of action potentials and promotes/stabilizes sleep. Previous failures to detect an α7-nAChR response in the TRN were almost certainly caused by electrical stimulation methods, i.e. ACh perfusion. Because α7-nAChRs desensitize so quickly (*Giniatullin et al., 2005*), their responses can be missed in experimental paradigms. Another group found that Excitatory Postsynaptic Potentials (EPSPs) induced by electrical stimulation were blocked by dihydro-β-erythroidine (DhβE, 300 nM), indicating that the EPSPs were mediated by α4β2-containing nAChRs (*Sun et al., 2013*). However, DhβE (300 nM) can block both α4β2- and α7 nAChR-mediated currents (*Alkondon and Albuquerque, 1993*), and non-specific electrical stimulation may enable extrasynaptic transmitter release.

Cholinergic neurons in the basal forebrain send projections to many targets, including the cerebral cortex, hippocampus, thalamus, and TRN. Selective activation of cholinergic neurons in the basal forebrain has been shown to induce an immediate transition from sleep to waking (*Han et al., 2014*). The present study specifically activated cholinergic fibers locally in the TRN, which, surprisingly, showed the opposite effect. It induced transition from waking to sleep, indicating that direct cholinergic projections to the TRN promote sleep onset and sleep stabilization. GABAergic neurons of the TRN fire AP bursts mediating slow and spindle waves (*Cain and Snutch, 2010*). These bursts are very important for the generation of thalamocortical spindle waves (*von Krosigk et al., 1993*; *Lee et al., 2013*), and spindle rhythms play important roles in protecting sleep initiation and elevating arousal threshold (*Steriade, 2005*; *Wimmer et al., 2012*). We propose that excitation of cholinergic fibers in the TRN acts on GABAergic neurons through α7-nAChRs (*Roth et al., 2000*) to depolarize the membrane sufficiently to induce bursts of APs. Our results demonstrate a significant positive relationship between spindle density and the promotion of the NREM state and sleep stabilization. Activation of the cholinergic neurons appears to play critical roles in the switch from waking to sleep and in the protection of sleep itself, especially by increasing the NREM state.

It is possible that stimulating axonal fibers may cause antidromic and collateral activation because TRN-projecting cholinergic neurons mainly involved in the nBM and PPTg also project to other brain areas. Cholinergic neurons in the nBM of the basal forebrain project to the cerebral cortex, hippocampus, and TRN, and that in the PPTg of the brainstem project to the thalamus, new striatum, and basal forebrain. Activation of cholinergic fibers in the TRN may antidromically collaterally excite cell bodies, or collaterally act on other neurons, leading to the indirect effects on sleep. To address this

question, we locally injected the α7 receptor blocker MLA in the TRN and found that sleep induced by cholinergic fiber stimulation was completely inhibited, suggesting that the possibility was small.

We found that unilateral stimulations and drug infusions had significant effects on behavior and phenotype. This did not exclude spontaneous behavior evoked by contralateral Ach release. Bilateral stimulation is better to clearly address the mechanisms.

The finding that optical stimulation of cholinergic projections to the TRN directly elicits EPSCs in the GABAergic neurons via α7-nAChRs is unexpected. Though α7-nAChRs are widely distributed in the brain, they are usually thought to provide a modulatory role. From presynaptic locations, the receptors can enhance transmitter release (*McGehee et al., 1995*; *Gray et al., 1996*), while from post- or presynaptic locations they can modulate synaptic responses indirectly through calcium-dependent mechanisms (*Dajas-Bailador and Wonnacott, 2004*). Rarely, if ever, have α7-nAChRs been shown to be the primary mediator of rapid, synaptic transmission in the central nervous system (CNS). In the present study, the ability of MLA to block cholinergically-induced EPSCs in the TRN clearly implicates α7-nAChRs. Immunostaining corroborated the presence of α7-nAChRs in the GABAergic neurons of the TRN. The inability of CNQX, AP5, and PTX to block the EPSCs demonstrated that rapid glutamatergic and GABAergic transmission were not responsible for the EPSCs. The fact that α7-nAChRs have a high relative permeability to calcium (*Bertrand et al., 1993*) and trigger calcium release from internal stores (*Dajas-Bailador and Wonnacott, 2004*) may be relevant for the kinds of signaling effective at promoting spindle oscillations.

Our study provides critical insight into cholinergic roles in mediating biological rhythms. Cholinergic neurons play an important role in waking-to-sleep cycles and in sleep stabilization by activating GABAergic neurons in the TRN through α7-nAChRs. Although a great deal of work has been done to examine the roles of cholinergic systems in biological rhythms, previous experiments have mainly focused on the sleep-to-waking transition, and results have been limited by the experimental techniques used. Cholinergic projections from different brain areas to the TRN may have opposing effects, acting through different receptors and synaptic pathways (*Everitt and Robbins, 1997*; *Abbott et al., 2003*). The states of sleep and waking may be determined by the opposing inputs serving as a system of checks and balances. Because a number of psychiatric disorders, such as schizophrenia (*Ferrarelli and Tononi, 2011*), show sleep disturbances and loss of spindle oscillation, the results reported here may also be useful in suggesting new targets for therapeutic intervention in these cases.

## Materials and methods

### Animals

Mice were housed in individual Plexiglass recording cages in temperature (24 ± 1°C) and humidity (40–60%) controlled recording chambers under a 12/12 hr light/dark cycle (starting at 08:00). Food and water were available *ad libitum*. *PV-Cre* mice were provided by Z.J. Huang (Cold Spring Harbor, USA) and X.H. Zhang (Beijing Normal University, China), and *ChAT-ChR2* and *VGAT-ChR2* mice were provided by G.P. Feng (Massachusetts Institute of Technology, USA) and M.M. Luo (National Institute of Biological Sciences, China). All experiments were carried out in accordance with the guidelines described in the National Institutes of Health Guide for the Care and Use of Laboratory Animals, and were approved by Committee of Laboratory Animal Center of Zhejiang University (ZJU201553001).

### Brain slice preparation and electrophysiological recordings

Mice were deeply anesthetized with isoflurane (1.5–3%), the brain was removed after decapitation, and slices (300 μm thick) were then cut in a solution containing (in mM): 250 sucrose, 26 $NaHCO_3$ 10 D(+)glucose, 4 $MgCl_2$, 2.5 KCl, 2 sodium pyruvate, 1.25 $H_2NaPO_4$, 0.5 ascorbic acid, 0.1 $CaCl_2$, and 1 kynurenic acid, pH 7.4. Recordings were made at 34°C (TC-324B; Warner) in artificial cerebrospinal fluid solution containing (in mM): 126 NaCl, 26 $NaHCO_3$, 1.25 $NaH2PO_4$, 3 KCl, 2 $CaCl_2$, 2 $MgCl_2$, and 10 D-glucose. All external solutions were saturated with 95% $O_2$/2.5% $CO_2$ gas. Whole-cell patch-clamp recordings were made with pipettes (3–4 MΩ) filled with internal solution containing (in mM): 130 K-gluconate, 2 NaCl, 4 $MgCl_2$, 20 Hepes, 4 $Na_2ATP$, 0.4 $Na_3GTP$, 1 EGTA, and 10 $Na_2$phosphocreatine, pH 7.25, with 1 M KOH; 290–300 mOsm. CNQX, AP5, and PTX were purchased from

Sigma. Recordings were obtained using the Axopatch 700B amplifier and Digidata 1440A (Molecular Devices), and neurons were visualized using infrared differential interference contrast visualization through a Nikon E600FN microscope and a CCD camera (Hamamatsu, Japan). Data were filtered at 10 kHz and digitized at 20 kHz, and acquired and analyzed using pCLAMP10.2 software (Molecular Devices). The laser was controlled by analog commands from the Digidata1440A (Molecular Devices).

## Immunohistochemistry

Eight-week old mice were perfused with 4% paraformaldehyde in PBS. The brains were harvested and placed in 30% sucrose solution for 24 hr, then embedded in OCT (optimal cutting temperature; Tissue-Tek; Sakura Finetek, USA) and subsequently cut into 30-µm thick slices. The slices were stained using monoclonal anti-parvalbumin (1:4500; clone parv-19; Sigma) and antibody rabbit anti-VAChT (Synaptic System; 1:500) as the primary antibodies. After several washes with PBS, sections were incubated with Alexa Fluor-488 goat anti-mouse IgG (Invitrogen). Fluorescence was detected using an Olympus confocal microscope and standard FITC filters.

## Virus construction and injection

For rabies virus-mediated retrograde monosynaptic neuronal tracing, several AAV backbone plasmids were constructed in our lab or ordered from Addgene. The pAAV-EF1a-FLEX-GT was acquired from E.d. Callaway's lab; pAAV-CAG-DIO-GT was constructed by sub-cloning the CAG promoter and the coding region of EGFP:2A:TVA from pAAV-EF1a-FLEX-GT (Addgene plasmid 26198). The pAAV-EF1a-DIO-RV-G was constructed by sub-cloning the CAG promoter and the coding region of RV-G from pAAV-EF1a-FLEX-GTB (Addgene plasmid 26197) into the pAAV-EF1a-double floxed-hChR2 (H134R)-EYFP plasmid (Addgene plasmid 20298). All rAAV viruses were packaged into serotype 9 with titer at approximately $1 \times 10^{13}$ genome copies per milliliter (Neuronbiotech). Two- to three-month-old *PV-Cre* mice were used for retrograde transsynaptic tracing. A volume of 0.2 µl of a 1:1 mixture of AAV9-CAG-DIO-EGFP-2A-TVA ($1.3 \times 10^{13}$ particles/ml) and AAV9-CAG-DIO-RV-G ($1.5 \times 10^{13}$ particles/ml) was stereotaxically injected into the TRN (coordinates in mm: AP -0.65, ML 1.30, DV -3.20) using a stereotaxic injector (Ruiwode Life Science, Shenzhen, China). Three weeks later, 0.5 µl of rabies-DsRed virus ($3 \times 10^{8}$ particles/ml) was injected into the same site.

## Animal surgery

For all experimental tests, 6- to 88-week-old male mice were anesthetized with 1.5–33% isoflurane and placed in a stereotaxic frame. For each animal, five stainless-steel screws were implanted in the skull to provide EEG contacts, ground, and mechanical support for the drive. Epidural electrodes were implanted in the frontal (anteroposterior, +2.0 mm; ML, -1.5 mm) and parietal lobes (anteroposterior, -2.0 mm; ML, -3.0 mm), and a grounding electrode was implanted in the occipital region. For EMG signal recording, two polyester enameled wire electrodes were inserted into the neck musculature to record postural tone. The implant was attached to a custom-designed stereotaxic arm and lowered to the following stereotaxic coordinates (M/L 1.7 mm, A/P 0.8 mm, D/V -3.2 mm). Insulated leads from the EEG and EMG electrodes were then soldered to a mini connector. For each animal, the cannula was secured using dental cement.

## In vivo recording

After one week of recovery, each mouse was connected to the A-M system 1800 amplifier. The amplified cortical EEG and EMG signals were acquired (sampling frequency, 2 kHz) by Digidata 1440A (Molecular Devices). The data were acquired as continuous 1-hr segments with different stimulation protocols. For *ChAT-ChR2* mice, the stimulation was persistent 200-ms laser pulses with an interval of 6 s. For *VGAT-ChR2* mice, the stimulation was 8 Hz, 10 ms pulses with an interval of 6 s. Simultaneous video monitoring was performed for the visual inspection of sleep behavior. All scoring was measured by combined EEG, EMG, and video recordings.

## Photostimulation

A 473 nm laser from a solid-state laser diode (Newdoon) was delivered to the TRN though an optical fiber (200 µm, NA = 0.22). Prior to connecting to the cannula, laser power was measured to 3 mW

using an optical power meter (Thorlabs, S121c). Based on measurement prior to surgery, we found 50% losses of laser power at the tip of the implanted fiber. The ultimate laser power was about 1.5 mW at the fiber tip. The 1440A was used to shape laser pulses of 5 or 10 ms duration at different frequencies.

### Spindle detection method

Sleep spindles were analyzed using Clampfit 10.2 (Axon Instruments) and NeuroExplorer4 (Plexon) in the wake and NREM sleep stages and detected using the following complementary method: the EEG time series were bandpass-filtered in the frequency range of 7–15 Hz, and signal was computed in moving windows of 1 s duration and at 0.01 s intervals. Sleep spindles were detected during those times in which the root mean square (rms) value of the filtered signal exceeded its 85th percentile, as previously suggested (*Dang-Vu et al., 2010*). Detected spindles with durations lower than 0.5 s or longer than 5 s were discarded. To estimate spindle density (SD) during a given sleep stage, all spindles were counted and the total number was divided by the total recording time.

### Power spectrum

Power spectrum analyses of sleep stages and spindles were performed in moving windows of 1 s duration and at 0.01 s intervals, and power spectrum distribution was estimated with a resolution of 0.5 Hz. The post power spectrum was smoothed by a 'Boxcar' filter with a width of 6.

### Statistical analysis

The data were collected and processed randomly. All immunocytochemistry and behavioral tests were carried out blind. The electrophysiology experiments were not blind, but data collection and analyses were performed blind. Sample size was calculated according to the preliminary experiment results and the following formula: $N = [\frac{(Z_{\alpha/2}+Z_{\beta})\,\sigma}{\delta}]^2(Q_1^{-1} + Q_2^{-1})$, where $\alpha$ = 0.05 significant level, $\beta$ = 0.2, power = 1-$\beta$, $\delta$ is the difference between the means of two samples, and Q is the sample fraction. The samples were randomly assigned to each group. The data were presented as mean ± SEM. One-way ANOVA was used for analysis of electrophysiological data in vitro. Two-tailed Student's *t*-tests were performed after the normality test for the data from all groups in vivo. Variance was similar between groups being compared. We considered $p < 0.05$ to be statistically significant. The sleep patterns were determined by combined EEG, EMG, and video recordings. Data were presented as mean ± SEM. All data were analyzed using Origin8.0 (OriginLab), Clampfit 10.2 (Axon Instruments), NeuroExplorer4 (Plexon), and Microsoft Excel 2010. Data were exported into Illustrator CS6 (Adobe Systems) for preparation of figures.

## Acknowledgements

We thank ZJ Huang (Cold Spring Harbor, USA) and XH Zhang (Beijing Normal University, China) for kindly providing the *PV-Cre* mice line, GP Feng (Massachusetts Institute of Technology, USA) and MM Luo (National Institute of Biological Sciences, China) for the *ChAT-ChR2-EYFP* mice line, and FQ Xu (Wuhan Institute of Physics and Mathematics, the Chinese Academy of Sciences, China) for kindly providing the virus. This work was supported by grants from the National Natural Science Foundation of China for Distinguished Young Scientists (81225007), the Key Project of the China National Natural Science Foundation (31430034), the Major Research Plan of the National Natural Science (91432306), and Funds for Creative Research Groups of China (81221003), Program for Changjiang Scholars and Innovative Research Team in University, and Fundamental Research Funds for the Central Universities. This work was also sponsored by the Zhejiang Province Program for Cultivation of High-level Health Talents.

## Additional information

### Funding

| Funder | Grant reference number | Author |
|---|---|---|
| National Natural Science Foundation of China | Distinguished Young Scientists 81225007 | Xiao-Ming Li |
| Key Project of the China National Natural Science Foundation | 31430034 | Xiao-Ming Li |
| Major Plan of the National Natural Science | 91132714 | Xiao-Ming Li |
| Science Fund for Creative Research Groups | 81221003 | Xiao-Ming Li |

The funders had no role in study design, data collection and interpretation, or the decision to submit the work for publication.

### Author contributions

K-MN, Conception and design, Acquisition of data, Analysis and interpretation of data, Drafting or revising the article, Contributed unpublished essential data or reagents; X-JH, Acquisition of data, Contributed unpublished essential data or reagents; C-HY, PJ, Acquisition of data; PD, Acquisition of data, Analysis and interpretation of data; YL, Acquisition of data, Drafting or revising the article; YZ, DKB, SD, Drafting or revising the article; X-ML, Conception and design, Analysis and interpretation of data, Drafting or revising the article, Contributed unpublished essential data or reagents

### Ethics

Animal experimentation: All experiments were carried out in accordance with the guidelines described in the National Institutes of Health Guide for the Care and Use of Laboratory Animals, and were approved by Committee of Laboratory Animal Center of Zhejiang University(ZJU201553001).

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
