## [Decision Letter]

Thank you for submitting your work entitled "Selectively Driving Cholinergic Fibers Optically in the Thalamic Reticular Nucleus Promotes Sleep" for consideration by *eLife*. Your article has been reviewed by two peer reviewers, and the evaluation has been overseen by a Reviewing Editor (Joseph Takahashi) and a Senior Editor.

The reviewers have discussed the reviews with one another and the Reviewing Editor has drafted this decision to help you prepare a revised submission.

Summary:

It is commonly believed that cholinergic neurons in the basal forebrain and brainstem drive arousal and promote attention. In this study, Ni et al. report their surprising finding that stimulating cholinergic inputs to the thalamic reticular nucleus (TRN) facilitates sleep onset and protects non-rapid eye movement (NREM) sleep, likely through increasing spindle oscillations of TRN neurons. In addition, cholinergic inputs directly evoke fast excitatory postsynaptic currents (EPSCs) and elicit bursts of spike firing by targeting α7-containing nicotinic acetylcholine receptors. These discoveries are significant because it challenges the dogma that cholinergic activity is positively associated with arousal. The finding of rapid cholinergic EPSCs is also novel because acetylcholine typically produces modulatory effects via volume transmission in the brain. The manuscript is well written and the figures are well organized.

Essential revisions:

1) The stimulation conditions are arbitrary; 200 ms pulse duration is significantly longer than the average and could be inducing artifactual firing. A concern is how the stimulation parameters are related to the physiological activity patterns in behaving animals. To stimulate cholinergic fibers, the authors used laser pulses of different durations (20, 100, 200, and 400 ms) and found that 200 ms-long pulses were most effective. To stimulate TRN neurons, they used 8 Hz 10-ms pulses. Although these types of stimulations produced clear behavioral effects, they do make a reader wonder how a 200 ms-long light pulse or an 8 Hz train of short light pulses changes the firing rates of TRN-projecting cholinergic neurons or TRN neurons, respectively.

2) In addition, it is unclear how the stimulation effects mimic neuron activity patterns before and during NREM sleep. Would it be possible that stimulating with a different parameter (tonic activation) actually promotes arousal? I understand that some of the questions are difficult to address experimentally. Nevertheless, the authors can test how the light pulses of different duration affect neuronal activity in brain slices and discuss the potential relevancy and caveats.

3) The origin of cholinergic fibers is still unknown. I would suggest using viral injections in *ChAT cre* animals to map the source of TRN innervation.

4) All sleep experiments should be accompanied by power spectral analysis to determine the quality of sleep induction.

5) The authors should cite papers or provide experimental data on the source of cholinergic inputs to the TRN. Do TRN-projecting cholinergic neurons also project to other brain areas? If so, they should discuss the possibility that stimulating axonal fibers may cause antidromic and collateral activation. The effect of the α7 blocker MLA partially addressed this concern.

6) All stimulations and drug infusion appear to be applied unilaterally. The authors should clearly describe and discuss this potential limitation.

7) Group data are needed for Figure 4. Moreover, this figure panel shows that 200 ms pulses elicited burst spike firing, which differs from the pattern of 8 Hz 10 ms pulses for TRN stimulation. This difference should be discussed.

8) The laser power was described either as 1.5 mW or 15 mW. The authors should check the accuracy of these two numbers.

9) In Figure 2 legend, "In (E, G, G)" should be "In (E, F, G)".

---

## [Author Response]

*Essential revisions: 1) The stimulation conditions are arbitrary; 200 ms pulse duration is significantly longer than the average and could be inducing artifactual firing. A concern is how the stimulation parameters are related to the physiological activity patterns in behaving animals. To stimulate cholinergic fibers, the authors used laser pulses of different durations (20, 100, 200, and 400 ms) and found that 200 ms-long pulses were most effective. To stimulate TRN neurons, they used 8 Hz 10-ms pulses. Although these types of stimulations produced clear behavioral effects, they do make a reader wonder how a 200 ms-long light pulse or an 8 Hz train of short light pulses changes the firing rates of TRN-projecting cholinergic neurons or TRN neurons, respectively.*

We are very sorry we did not describe this point clearly in the original version of the manuscript. Thank you for the suggestions.

In effect, we stimulated cholinergic neurons in the nBM and found that a 5 ms pulse could induce the firing of action potentials in cholinergic neurons (see revised Figure 4). However, both 5 ms and 50 ms pulse stimulation of cholinergic terminals failed to induce bursts of action potentials and spindles in GABAergic neurons, with at least 100 ms required to induce bursts of action potentials and spindles related to NREM sleep duration (revised Figure 4 and Figure 1). We believe that the activation of the GABAergic neurons by cholinergic fiber stimulation might be an indirect effect via α7 cholinergic receptors (Figure 5). It requires a certain amount of Ach release to excite postsynaptic GABAergic neurons to fire action potentials and spindles. Light pulses of 5 ms or 50 ms are too short to induce the release of sufficient Ach, which evokes the burst of action potentials in postsynaptic GABAergic neurons and spindles.

However, light-stimulation of GABAergic neurons in the TRN is direct, and only a short pulse of 10 ms (or even shorter) induced the burst of action potentials. The use of 8 Hz mimicked the frequency of spindles at which spindle induction was most effective.

*2) In addition, it is unclear how the stimulation effects mimic neuron activity patterns before and during NREM sleep. Would it be possible that stimulating with a different parameter (tonic activation) actually promotes arousal? I understand that some of the questions are difficult to address experimentally. Nevertheless, the authors can test how the light pulses of different duration affect neuronal activity in brain slices and discuss the potential relevancy and caveats.*

Thank you for your suggestion. We tried 5 ms, 50 ms, 100 ms, and 200 ms light pulses to stimulate the cholinergic fibers, and recorded neuronal activity of GABAergic neurons in the TRN (revised Figure 4). We found that short duration light pulses, such as 5 ms and 50 ms, failed to induce bursts of action potentials in GABAergic neurons. A light pulse of at least 100 ms duration was required to induce bursts of action potentials and spindles. We believe that the activation of GABAergic neurons by cholinergic fiber stimulation might be an indirect effect via the α7 cholinergic receptors (Figure 5). It requires a certain amount of Ach release to excite postsynaptic GABAergic neurons to fire action potentials and spindles. Light pulses of 5 ms or 50 ms are too short to induce the release of sufficient Ach, which evokes the burst of action potentials in postsynaptic GABAergic neurons and spindles.

Spindles induced by a 200 ms pulse were very similar to natural, in both frequency and amplitude (Figure 1). More importantly, the relevancy of spindle density and NREM duration was consistent with that of the natural ones (Figure 3, *ChAT-ChR2*-NonSti).

Whether tonic activation promotes arousal is a very good question. We will investigate this and mechanisms in our next project, and hope the reviewers understand that we could not address this problem in the current research.

*3) The origin of cholinergic fibers is still unknown. I would suggest using viral injections in ChAT cre animals to map the source of TRN innervation.*

Thank you for this suggestion. We injected the retrograde monosynaptic rabies virus with DsRed (please see revised methods) in PV-positive neurons of the TRN, with the DsRed signal then detected in cholinergic neurons of the nBM of the basal forebrain (Figure 4—figure supplement 1) and of the PPTg of the brainstem (Figure 4—figure supplement 1). Statistical results showed that ~30% of cholinergic neurons (VAChT antibody labeled) were positive for mCherry (retrograde virus labeled) in the nBM, and 7% were positive for mCherry in the PPTg, suggesting that the TRN receives cholinergic projections from both the nBM and PPTg.

*4) All sleep experiments should be accompanied by power spectral analysis to determine the quality of sleep induction.*

Thank you; the manuscript has been revised according to this suggestion.

*5) The authors should cite papers or provide experimental data on the source of cholinergic inputs to the TRN. Do TRN-projecting cholinergic neurons also project to other brain areas? If so, they should discuss the possibility that stimulating axonal fibers may cause antidromic and collateral activation. The effect of the α7 blocker MLA partially addressed this concern.*

This is good suggestion, and related papers have been cited (see References). We also performed new experiments and found that the TRN receives cholinergic projections from both the nBM and PPTg (please see Figure 4—figure supplement 1).

We agree with the reviewers. It is possible that stimulating axonal fibers may cause antidromic and collateral activation because TRN-projecting cholinergic neurons mainly involved in the nBM and PPN also project to other brain areas. Cholinergic neurons in the nBM of the basal forebrain project to the cerebral cortex, hippocampus, and TRN, and that in the PPN of the brainstem project to the thalamus, new striatum, and basal forebrain. Activation of cholinergic fibers in the TRN may antidromically collaterally excite cell bodies, or collaterally act on other neurons, leading to the indirect effects on sleep. To address this question, we locally injected the α7 receptor blocker MLA in the TRN and found that sleep induced by stimulation of cholinergic fibers was completely inhibited, suggesting that the possibility was small.

The manuscript has been revised according to reviewers’ suggestion. Please see the Discussion section of the revised manuscript.

*6) All stimulations and drug infusion appear to be applied unilaterally. The authors should clearly describe and discuss this potential limitation.*

Thank you for this very good suggestion. In effect, we found that unilateral stimulations and drug infusions had significant effects on behaviors and phenotypes. However, these may not exclude the spontaneous effects evoked by contralateral Ach release. Bilateral stimulation was better to clearly address the mechanisms, and has been discussed in the revised manuscript.

*7) Group data are needed for Figure 4. Moreover, this figure panel shows that 200 ms pulses elicited burst spike firing, which differs from the pattern of 8 Hz 10 ms pulses for TRN stimulation. This difference should be discussed.*

We have updated the original data with groups in revised Figure 4. When stimulating cholinergic fibers in the TRN, a 200 ms pulse was required to induce bursts of action potentials in GABAergic neurons. However, only a short pulse (such as 5 ms) induced the burst of action potentials in GABAergic neurons in the TRN when directly stimulating GABAergic neurons. In fact, both a short pulse (such as 20 ms) (Michael M Hlassa, et al., Nature Neuroscience, 2011) and short pulses (8 Hz 10 ms) (Angela Kim, et al., PNAS, 2012) can induce spindles. Here, we used 8 Hz for the purpose of inducing spindles most effectively. This difference has been clearly addressed in the new Figure 4 of the revised manuscript.

*8) The laser power was described either as 1.5 mW or 15 mW. The authors should check the accuracy of these two numbers.*

Sorry, this was a typing error. “15 mW” should be “1.5 mW”.

9) In Figure 2 legend, "In (E, G, G)" should be "In (E, F, G)".

Sorry, this was a typing error. Corrections have been made in the revised version.